# Flexible Work: Opportunity and Challenge (FLOC) for individual, social and economic sustainability. Protocol for a prospective cohort study of non-standard employment and flexible work arrangements in Sweden

Sven Svensson [ID],[1] David M Hallman [ID],[1] SvendErik Mathiassen,[1] Marina Heiden,[1] Arne Fagerström,[2] Jean Claude Mutiganda,[2] Gunnar Bergström[1,3]

For numbered affiliations see end of article.

**Correspondence to**
Dr Sven Svensson;
sven.svensson@hig.se

## ABSTRACT

**Introduction** Flexibility in working life, including non-standard employment (NSE) and flexible work arrangements (FWAs), offers the organisation a better ability to adapt to changing conditions while also posing considerable challenges for organisations as well as workers. The aim of the Flexible Work: Opportunity and Challenge (FLOC) study is to investigate associations between NSE and FWA on the one hand, and individual, social and economic sustainability on the other.

**Methods and analysis** This prospective open cohort study targets approximately 8000 workers 18–65 years old in 8–10 public and private organisations in Sweden. We will use a comprehensive battery of measurement methods addressing financial performance, physical and psychosocial exposures, and physical and mental health, both at the organisational and the individual level. Methods include valid survey questionnaires and register data, and, in subpopulations, technical measurements, interviews and diaries. Main exposures are type of employment and type of work arrangement. Main outcomes are indicators of social and economic sustainability and, at the individual level, health and well-being. Data, collected over 54 months at approximately 18-month intervals, will be analysed using multivariate methods considering main effects as well as potential effect modifiers. The analyses will take into account that respondents are nested in organisations, divisions and/or have specific managers.

**Ethics and dissemination** FLOC is approved by the Swedish Ethical Review Authority (decision numbers 2019–06220, 2020–06094 and 2021–02725). Data will be published in peer-reviewed journals and presented at international conferences, and researchers will assist the organisations in improving policies and routines for employment and organisation of work.

## STRENGTHS AND LIMITATIONS OF THIS STUDY

⇒ Recruiting organisations in different sectors employing workers with different socioeconomic status, ethnicity and gender allows for addressing effect modification in subgroups with varying employment conditions and work arrangements.
⇒ Repeated measurements of exposures and outcomes allow for estimation of temporal relationships.
⇒ Flexible Work: Opportunity and Challenge will combine survey data, interviews and diaries with objective data collected from company records, registers and technical measurements of physical workload and psychophysiological indicators. This will offer a comprehensive array of exposures and outcomes.
⇒ Non-standard workers are only temporarily affiliated with a certain employer. This could result in a high dropout rate.

## INTRODUCTION

Increased global competition and new information and communication technology has brought about an increased need as well as an improved capacity for flexibility in organisations.[1] Non-standard employment (NSE, the hiring of different sorts of temporary staff)[2] mainly provides flexibility for the organisation, whereas flexible work arrangements (FWAs, employment agreements allowing work to be performed at a discretionary time and place) mainly provide flexibility for workers.[3]

Both NSE and FWA could potentially benefit both employers and workers. However, this potential is not always realised; NSE and FWA can imply more autonomy but also that you will have to follow others' needs.[4 5] This is highly actualised during the ongoing COVID-19 pandemic, where many organisations have been able to maintain operations and employments because of FWA. At the same time, non-standard workers (NSWs) have in many cases been the first to

be laid off.[6] Thus, understanding the trade-off between positive and negative aspects of flexibility in working life is a pertinent issue.

NSE and FWA may also represent challenges regarding work environment and health challenges that can hamper social and economic sustainability. Social and economic sustainability refers to the long-term balance between the needs of different stakeholders in organisations and companies. This includes organisations' need to maintain sound finances while at the same time providing a health-promoting work environment[7 8] and even securing the needs of individuals for sustainable health and well-being until retirement.[8]

Results regarding organisational effects of NSE and FWA are rare and mixed. Some evidence suggests that industries with a large extent of NSE have a weaker product innovation propensity[9] and that NSE is associated with either higher or lower wage costs.[10] Both issues are matters of current debates.[9 10] Literature addressing effects of hiring NSW on the economic performance of an organisation is sparse, although low as well as high proportions of NSE might be negatively associated with financial performance.[11]

### Work environment and health in NSE

It may be difficult to involve NSW in occupational health and safety routines, for example, due to ambiguities in the division of responsibilities[12] or communication difficulties.[13] NSE appears associated with physical and mental ill health[14–16] and increased risk of occupational injury.[17] However, mixed findings have been reported.[15 17 18] NSE often results in poor physical and psychosocial work environments.[14 19–22] However, differences compared with standard work appear small.[23]

Studies of NSE often suffer from methodological weaknesses.[15 17 18 24] Specific types of NSE appear to be differently associated with work environment and health, but most studies mix various forms of NSE.[24] Most studies are cross-sectional. Few studies have compared the working conditions of NSW to those of standard workers. Most previous studies have focused on the psychosocial rather than the physical work environment. Most often, non-validated instruments and self-reports have been used,[14 18] likely introducing bias, for example, when measuring physical activity.[25]

### Work environment and health in flexible work

Before COVID-19, 40%–65% of the Swedish workforce could, to various extents, decide their working hours[26] or work from home.[27] The prevalence rates of telework (eg, working from home) have increased since the outbreak of the COVID-19 pandemic.[28 29] Of those working from home in the summer of 2020 due to COVID-19, 46% did so for the first time.[30]

FWA may improve work–life balance for workers and potentially benefit the productivity of organisations.[31] At the same time, employers may find it difficult to maintain occupational health and safety routines and

a good psychosocial working environment for flexible workers.[32–34] The proportion of FWA in an organisation may influence the working conditions even for workers with non-FWA.[35] Although FWA may allow workers a better work–life balance,[36] it also puts higher demands on workers to set limits to their work.[37 38] Failure to set such limits may contribute to excessive workload and overtime work, insufficient time for recovery and increased stress.[39] Most research on health in workers with FWA is cross-sectional,[31 36] and evidence of long-term health effects of FWA remains scarce.[40] Prospective studies are needed to improve knowledge, aiming at determining potential temporal associations between NSE and FWA on the one hand, and individual, social and economic sustainability on the other.[41–44]

### Overall aim

The overall aim of the present Flexible Work: Opportunity and Challenge (FLOC) study is to determine in which ways and to which extent NSEs and FWAs are associated with social and economic sustainability, in terms of the financial performance of organisations, their work environment and workers' health and well-being.

#### Primary aims and research questions (RQs)

#### Aim 1

The first objective of this study was to evaluate the extent to which different forms of NSEs and FWAs are associated with the financial performance of the organisation, its work environment, and health and well-being among the workers.

> *RQ1*: What are the opportunities and challenges presented by NSE/FWA in the context of an organisation's financial performance and its management accounting and control?
> *RQ2*: To which extent are different forms of NSE/FWA related to physical and psychosocial exposures at work?
> *RQ3*: To which extent are different forms of NSE/FWA related to outcomes describing health and well-being?
> *RQ4*: To which extent do the answers to RQ1–RQ3 differ between occupational sectors and according to age, gender and socioeconomic status of the worker?

#### Aim 2

The second objective of this study was to evaluate the contents and implementation of organisational policies and routines regarding NSE/FWA.

> *RQ5*: What are the contents of organisational NSE/FWA policies and routines?
> *RQ6*: How knowledgeable are managers, workers and union representatives regarding policies and routines for NSE and FWA?
> *RQ7*: How do managers use/implement policies and routines for NSE and FWA in everyday work?

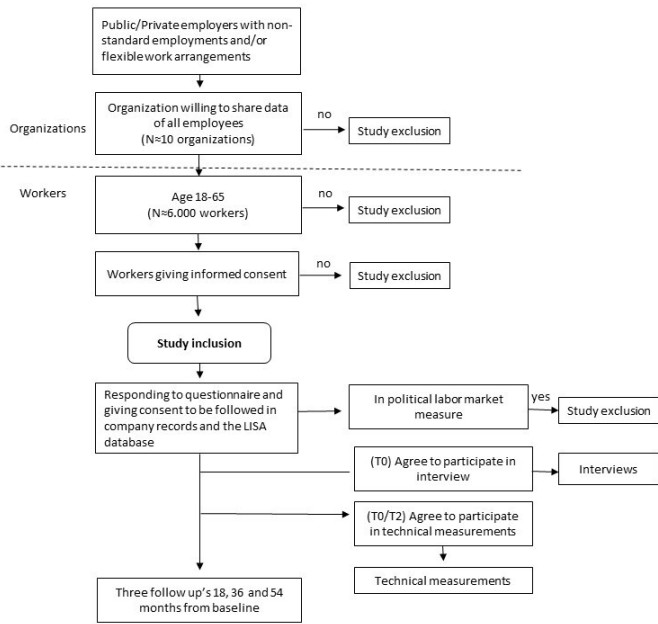

**Figure 1** Recruitment of study population and data collection (intended numbers). LISA, Longitudinal Integrated Database for Health Insurance and Labour Market Studies.

## METHODS AND ANALYSIS

### Design

FLOC is a prospective study with an open cohort approach; that is, new participants can be added at different time points. The cohort combines surveys with register data, technical measurements, diaries and interviews. Measurements are performed at four time points, about 18 months apart. The overall design and timeline of the study are shown in figures 1 and 2). The project is followed by a reference group with representatives from trade unions, the Confederation of Swedish Enterprise, the Swedish Work Environment Authority and universities.

### Recruitment

We will apply a purposive convenience sampling of organisations and employees within organisations. Recruitment strategies include distributing messages on a well-attended work environment blog, e-mailing potential organisations, and giving presentations for unions and employer organisations. The study population will, eventually, consist of 8–10 organisations within the private and

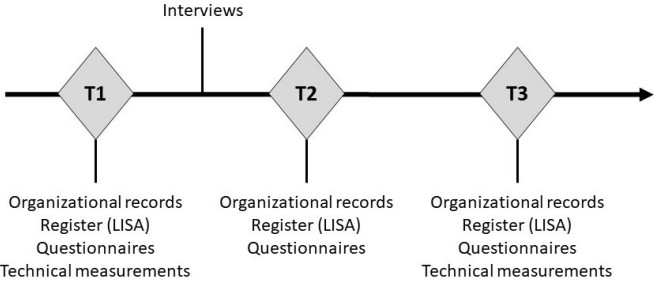

**Figure 2** Flow of the data collection. LISA, Longitudinal Integrated Database for Health Insurance and Labour Market Studies.

public sectors in Sweden, representative of specific types and extents of NSE and/or FWA.

### Eligibility criteria

Recruitment of organisations will be concentrated to mid-Sweden. We strive to include organisations from both the private and public sectors. Large organisations (>250) are prioritised. Eligible organisations will allow us to approach all employees, register FWA for all participating workers and/or have a relatively large fraction of NSWs. In total, the participating organisations will employ approximately 8000 blue-collar and white-collar workers. Eligible study participants are individuals 18–65 years old at baseline. Those employed because of political labour market initiatives at baseline are excluded (see figure 1). The organisations should be willing to share financial records as well as data on, for example, sick leave and form of employment, to allow measurements during working hours and to commit to the collaboration over several years.

### Study procedures

Data will be collected from approximately 8000 employees in the participating organisations between 25 May 2020 and 30 November 2025. Data include excerpts from the Longitudinal Integrated Database for Health Insurance and Labour Market Studies (LISA) at Statistics Sweden,[45] organisational data, a questionnaire to the workers, technical measurements of physical activities and heart rate, diaries and interviews. The questionnaire will be administered at baseline (T0), followed by three follow-ups at approximately 18 months intervals. Data will be combined with information from organisational records and the LISA register over the 4.5 year follow-up period (T0–T3). At each measurement point, participants will be asked for their informed consent to participate and will be followed via their social security number in the LISA database and in organisational records. Thus, follow-up data will be available even for study participants who drop out of the study during the data collection period. Participants will also be asked for their private contact information, which allows follow-up data to be collected from study participants who have, for instance, changed employer, become unemployed or turned to studies.

The FLOC questionnaire will mainly be composed of digital versions of validated questions from, for example, Copenhagen Psychosocial Questionnaire III (COPSOQ III),[46] the standardised questionnaire for the analysis of musculoskeletal symptoms,[47] and the Karolinska Sleep Questionaire[48] (see table 1 for a complete list of questionnaires). It will be sent to all employees in the participating organisations, primarily via e-mail. The electronic FLOC questionnaire will be hosted by Qualtrics (Provo, Utah, USA), with survey data stored at servers in the European Union. Depending on the accessibility of different groups of workers, the questionnaire will also be distributed via short message service (sms) or in a paper-and-pencil version.

**Table 1** Exposures, outcomes and contextual factors

| Primary exposures | |
|---|---|
| Type of employment contract | 1, 2a |
| Duration of employment contract | 1, 2a |
| Type of flexible work arrangement | 1, 2a |
| Extent of flexible work | 2a |
| Experience of flexible work | 2a |
| **Primary outcomes** | |
| Financial performance | |
| Value added and ratios | 1 |
| Sick leave | 1, 2b, 3 |
| Wages | 1, 3 |
| Turnover | 1, 3 |
| Productivity | 1, 2c, 2d |
| Presenteeism | 2b |
| Management control | 4 |
| Worker health and well-being | |
| General health | 2e |
| Musculoskeletal pain | 2f |
| Depression | 2g |
| Well-being | 2h |
| Job satisfaction | 2g |
| Work-ability | 2i |
| Exhaustion | 2g |
| Stress | 2j |
| Sleep quality | 2k |
| Need for recovery | 2l |
| Work–life balance | 2m |
| Work–life conflict | 2n |
| **Secondary exposures, secondary outcomes and contextual factors** | |
| Voluntariness, employment contract | 2a |
| Voluntariness and flexible work | 2a |
| Management and leadership quality | 2g, 4 |
| Job demands | 2g |
| Influence | 2g |
| Social support | 2g |
| Recognition | 2g |
| Predictability | 2g |
| Job insecurity | 2g |
| Vertical trust | 2g |
| Organisational justice | 2g |
| Offensive treatment | 2g |

Continued

**Table 1** Continued

| | |
|---|---|
| Psychological safety climate | 2o |
| Functionality, IT tools | 2a |
| Overtime work | 2a |
| Perceived demands for availability | 2g |
| Perceived flexibility | 2a |
| Occurrence/content of NSE policy | 1 |
| Occurrence/content of flexible work policy | 1 |
| Private expenses related to FW | 2a |
| Biomechanical load | 2p |
| General metabolic load | 5 |
| Physical activities during work and non-work time | 5 |
| Physical variation | 2g, 5 |
| Physical exertion | 2q |
| Mental exertion | 2a |
| Heart rate | 5 |
| General metabolic workload | 5 |

1, organisational records; 2a, self-developed survey item; 2b, survey[73]; 2c, survey[51]; 2d, survey[52]; 2e, survey[54]; 2f, survey[47]; 2g, survey[46]; 2h, survey[55]; 2i, survey[57]; 2j, survey[60]; 2k, survey[48]; 2l, survey[56]; 2m, survey[58]; 2n, survey[59]; 2o, survey[64]; 2p, survey[61]; 2q, survey[63]; 3, LISA register; 4, interview; 5, technical measurement. FW, flexible work; IT, information technology; LISA, Longitudinal Integrated Database for Health Insurance and Labour Market Studies; NSE, non-standard employment.

Data from organisational records regarding for example, types of employment or employment rates will be handed over by the organisations' human resources (HR) departments in Excel spreadsheets before the measurements, completed for any new worker at each follow-up. For participants having given their consent to be followed up in organisational records, the organisations will give complementing information regarding social security number, salary and annual days of sick leave for each employee. Salary and sick-leave data will be updated on a yearly basis. The social security numbers obtained from the organisational records will also be used to follow each employee and obtain outcome variables from the LISA register during 2020–2024.

Indicators for calculating organisational financial performance will be given by each organisation's accounts department and HR department in Excel spreadsheets during each data collection wave. These data will include information on organisational structure, including which employees are subordinate to specific managers in each organisational department. Documents for policies and routines will be provided by the HR departments.

In a subcohort, technical measurements of physical workload will be performed during work time and non-work time at T0 and T2 (details as follows). Participants will be recruited via an item in the questionnaire as well as via information given on notice boards in each organisation. Measurement equipment will be administered at the workplace or sent to the participant's home address along with an extensive instruction for attachment and operation. In the latter instance, a trained instructor is available via the video conference tool Zoom.

Interviews with managers will be performed after the baseline measurement. Participants will be recruited via the HR departments at the organisations, and via e-mail. The interviews will be recorded and transcribed verbatim.

In some cases, we will have access to two data sources, for example, as regards organisational policies. This allows us to examine possible differences between the two datasets and discuss these differences and their interpretation with relevant parties.

### Exposures, outcomes and contextual factors

Addressing the influence of NSE and FWA on social and economic sustainability at the organisational level and health and well-being at the individual level requires a number of exposures and outcomes to be followed. Also, a broad range of potential precursors and intermediate outcomes will be assessed to capture relevant aspects of physical and psychosocial exposures at work in different data sources. In addition, there is a need to consider characteristics and contextual factors specific to each organisation. Table 1 shows a complete list of exposures, outcomes, precursors and contextual factors, with references to data sources.

### Primary exposures

Main exposures are NSE and FWA. In the present study, NSE refers to different kinds of temporary employment and temporary agency workers.[2] The main FWA will be telework, or combinations of telework and on-site work, so-called hybrid work.[49] Data on NSE and FWA will be collected in the questionnaire and from organisational records. Self-reported exposure data will be used when no organisational data are available. NSE is recorded using a self-developed item based on the forms of employment registered in Statistic Sweden's Labour Force Surveys.[50] An FWA, according to the worker, is recorded using self-developed items for FWAs concerning its type, prevalence, duration and extent (days per week and total hours).

### Primary outcomes

Organisational records provide information about the financial performance of organisations, including indicators for value added, financial ratios, wages, turnover, productivity, sick-leave costs and—conditional on participants' consent—individual sick leave in days. The LISA register will be used to obtain additional outcome data of relevance for financial performance in the organisations for 2020–2024. Partial sick-leave days are transformed and standardised into full sick-leave days (net days); wages will be analysed at the individual level in terms of net incomes from employment after income tax as well as gross income before tax, and job turnover is measured as the number of employers during the follow-up period. Work performance at the individual level is measured by two items for productivity impairment, one due to ill health and one due to work environment problems.[51 52]

Self-rated sickness absence and presenteeism will be measured by one item each.[53] Individual semistructured interviews with 15–20 managers will give information on how management perceives control in the context of NSE and FWA. The interviews will concern managers' experiences of leading working groups with NSE or FWA, and their ideas of challenges and opportunities in connection with hiring NSW or in connection with FWA. Examples of question areas are advantages and disadvantages for managerial control of the working environment, and production consequences of hiring NSW or in connection with FWA.

Worker health and well-being will be self-rated, measured with questionnaires for general health (Short Form Health Survey, SF-36),[54] musculoskeletal pain,[47] exhaustion, depression and job satisfaction from COPSOQ III,[46] sleep quality (Karolinska Sleep Questionaire),[48] well-being (the WHO's well-being index, WHO-5),[55] need for recovery (short form scale),[56] workability (a single item from Ilmarinen[57]), work–life balance (a validated single item),[58] work–life conflict[59] and a valid stress item.[60]

### Secondary exposures, secondary outcomes and contextual factors

Information on working time is obtained through organisational records and in the questionnaire. Self-reported data regarding the physical and psychosocial work environment will include biomechanical load based on scales from the MUSIC study,[61] physical variation assessed with an item from COPSOQ III,[46] physical activity measured with the international physical activity questionnaire (IPAQ),[62] and physical exertion using a single item from Borg and Kaijser.[63] Additional self-developed items will be used to measure mental exertion and working postures. Psychosocial exposures (job demands, influence, social support, recognition, predictability, job insecurity, management quality, vertical trust, organisational justice, offensive treatment, demands for availability) is measured using indices or single items from COPSOQ III.[46] Psychosocial safety climate is measured with the four-item Psychosocial Safety Climate scale.[64] Self-developed questions address even functionality of information technology tools, overtime work, voluntariness regarding employment form and FWA, perceived demands for availability, perceived flexibility, occurrence of a policy for flexible work, commuting routines and private financial expenses related to flexible work.

Objective data on physical exposures during work and non-work time will be collected by technical measurements using small-size triaxial accelerometers fixed to the

thigh. Data will be collected over 7 days and processed and analysed using Acti4.[65] We will quantify the occurrence and temporal pattern of sitting, standing and physical activities.[66–68] Data will be allocated to work and non-work time based on diaries.[69] Heart rate will be measured using a Bodyguard2 monitor with a standard two-lead configuration, estimating general metabolic workload, for example, in per cent of heart rate reserve. Heart rate variability will be analysed[70] and used as a marker of autonomical cardiac modulation.

Organisational records will give information on availability and content of policies for NSE and FWA. They will also be used to nest respondents in organisational divisions and to specific managers. Interviews with managers, workers and union representatives will offer information on the knowledge of NSE/FWA policies and of the extent to which they are implemented.

### Confounders

To avoid the risk of mixing effects of a certain exposure with the effect of another variable, that is, confounding,[71] potential confounders will be identified from a large array of variables, for instance, demographic and socio-economic factors (including family situation), union membership, smoking habits, physical activity during leisure or prevalence of diagnosed disease. The selection of confounders will depend on the specific RQs in different studies, as well as theoretical and empirical considerations. Data on confounders will be collected from organisational records, the LISA register and the questionnaire. Data regarding the economic situation of the local community are available from Statistics Sweden.

### Sample size

The sample size was guided by a priori power analyses of primary exposures on the individual level (NSE/FWA) and outcomes (well-being, exhaustion, register information from organisational records on sickness absence in days). For minimally important score differences of 0.5 SD, we estimated that about 200 workers per employment category (eg, temporary contract and standard employment) or work arrangements category (eg, telework and on-site work) would be sufficient to detect significant ($p<0.05$) differences of 10 scale points with 80% power for well-being and exhaustion and 12 scale points for burnout.[46] A similar sample size is also sufficient to detect a difference of 20 days of sick leave between categories of workers. We aim at a target population of approximately 8000 workers in the cohort. Out of these, approximately 4000 will be standard workers, 2000 NSW and 2000 will have FWAs. In some cases, a particular worker may be represented in both the latter categories. We expect a response rate of approximately 50% at baseline and a total dropout of, at the most, 30%, which will give approximately 2800 workers in the final sample. This will give enough respondents to adjust for relevant covariates and possible effect modification (eg, by age and sex), in comparisons between categories of workers. Caution will

be taken to limit the number of variables in the models, as allowed by the size of the dataset.

### Data analysis

Workers with NSE will be compared with standard workers. Workers with FWA will be compared with those not having FWA. For FWA, duration and extent will be included as modifiers in the analyses. For both NSE and FWA, preference for NSE/FWA will be included as modifiers. Differences in outcomes over time will be analysed primarily using linear mixed models or generalised linear mixed models, depending on scale (continuous/categorical) and the distribution of data, considering main effects as well as potential modifiers. Respondents nested in different sectors, organisations and/or departments as well as to specific managers will be addressed specifically where possible. Groups may also be stratified based on gender, age and socioeconomic status. Theoretically relevant confounders will be considered in all analyses. Data loss will be inspected and treated using appropriate analytical imputation methods. Interviews will be analysed using thematic analysis, for example, qualitative content analysis.[72]

### Patient and public involvement

No patients were involved.

### ETHICS AND DISSEMINATION

The study is approved by the Swedish Ethical Review Authority (decisions numbers 2019–06220, 2020–06094 and 2021–02725), and will be performed according to the declaration of Helsinki. Participants answering the questionnaire will provide their informed consent in the online questionnaire. Participants in interviews and/or technical measurements will give their written informed consent to the researcher before starting the interview and/or starting measurements. FLOC data will be published in peer-reviewed journals and presented at international conferences. Participating organisations will not receive information that can be tracked down to individual workers. Reports of results at an aggregated level will be delivered to, and discussed with, participating organisations. Based on these discussions, researchers will assist the organisations in improvement of policies and routines, where relevant. The researchers will not receive any economic compensation from any participating organisation. Results from the study will be continuously discussed with the cohort's reference group, which consists of members from employers' organisations and unions, representatives for the Swedish Work Environment Authority and researchers. The FLOC project group will be open for collaboration with other researchers and data will be shared, conditional on the provision of a research plan to the project group and congruence with the ethical approval.

The FLOC study is expected to contribute with original evidence concerning the effects of flexible employment

forms and working arrangements. The study will initiate a new Swedish cohort on working conditions and health of workers with NSE and FWA. The study is based on a prospective approach with repeated measurements of exposures and outcomes, which allows for estimation of reliable risk estimates and minimises bias due to reversed causation. To the best of our knowledge, this is the first study that investigates both the physical and psychosocial working environments and its associated effect on financial performance in the organisation and on health among the workers in an NSE and FWA context. The combination of register data, organisational records, questionnaire data, technical measurements and interviews will give a comprehensive contribution to research addressing NSE and FWA. The combined focus on different forms of NSE and FWA also facilitates increased understanding of how increased flexibility in working life affects companies and employees. Enrolling large representable organisations in different occupational sectors that employ workers with different socioeconomic status, ethnicity and gender allows for addressing effect modification by employment conditions, work arrangements and personal factors. At the same time, the heterogeneity of organisations is a limitation of the project since it leads to reduced opportunities to generalise the results to a specific sector or type of organisational context. Nevertheless, the comprehensive set of variables studied, obtained from a variety of data sources, will enable us to draw conclusions of relevance to today's working life. The development towards increased flexibility in working life opens up challenges and opportunities for employers and employees alike. We hope that the FLOC study will contribute to a more sustainable working life for both employers and employees.

**Author affiliations**
[1]Department of Occupational Health Science and Psychology, University of Gävle, Gavle, Sweden
[2]Faculty of Education and Business Studies, University of Gävle, Gavle, Sweden
[3]Unit of Intervention and Implementation Research for Worker Health, Institute of Environmental Medicine, Karolinska Institutet, Stockholm, Sweden

**Contributors** GB, SEM, DMH, MH, AF and SS conceptualised the idea. GB, SEM, DMH, MH and SS conceived the study. GB, DMH, MH and SEM developed the overall design. JCM contributed to the section on economic and social sustainability. SS drafted the manuscript and all authors revised it critically. All authors have read and approved the manuscript in its present form.

**Funding** This work was supported by FORTE (grant number 2019–01257) and AFA Insurance (grant number 200244).

**Competing interests** None declared.

**Patient and public involvement** Patients and/or the public were not involved in the design, conduct, reporting or dissemination plans of this research.

**Patient consent for publication** Not applicable.

**Provenance and peer review** Not commissioned; externally peer reviewed.

**ORCID iDs**
Sven Svensson http://orcid.org/0000-0002-7798-1981
David M Hallman http://orcid.org/0000-0002-2741-1868

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
