## [Reviewer comments · BMJ Open]

ARTICLE DETAILS

TITLE (PROVISIONAL)	FLexible work: Opportunity and Challenge (FLOC) for individual, social and economic sustainability. Protocol for a prospective cohort study of non-standard employment and flexible work arrangements in Sweden
AUTHORS	Svensson, Sven; Hallman, David; Mathiassen, Svend; Heiden, Marina; Fagerström, Arne; Mutiganda, Jean Claude; Bergström, Gunnar

VERSION 1 – REVIEW

REVIEWER	McQuaid, Ronald University of Stirling, Management Work and Organisation
REVIEW RETURNED	11-Dec-2021

GENERAL COMMENTS	There are a number of issues that need to be resolved or clarified in the paper. The project appears to be ready so my comments related mainly to clarifications. The expected/actual start date should be clarified. General: This project should add useful research to important areas of understanding. It is important that sufficient resources can be applied to fully achieve the objectives and try to unpack the many confounding influences on employment outcomes. The expected statistical analysis is outlined in broad terms only (e.g. page 12) and so is difficult to assess at this stage. The literature is generally appropriate although there is a considerable literature in the social sciences (Economics, Sociology etc.) that would be relevant. There is considerable policy discussion about ‘informal employment’ (the informal economy is argued to be regulated, but by informal rather than formal institution, e.g. Williams, 2014) that might be worth mentioning, as some issues may be relevant, although it partly refers to work without contracts, that is excluded from this study (policy reports include ILO reports, European Union’s The European Pillar of Social Rights etc.). There are a number of points of clarification that would improve the Protocol. For clarity, I have usually tried to follow the order of the paper:
--

	Specific managers, and even organizational divisions, may change during the life of the project, so clarify the contingency plans for such events. What are dimensions of organizational sustainability and what effects it (which may vary by the sector it is in)? Clarification is needed on the differences between organizational 'sustainability' (e.g. page 5 line 24) and 'viability' (page 6 line 6) and 'financial viability' (page 6 line 13) and more generally what is meant by 'organizational sustainability'. For example, how do they differ? Also the issue of sustainability for NSE should be clarified. References should be given to link and expand on the section including "aiming at determining potential temporal associations between NSE and FWA on the one hand, and organizational viability, the work environment, and health and well-being of the workers, on the other." (page 6 lines 6/7). In terms of the aims, it could be made explicit that NSE and FWA may affect different workers but might affect the same worker (as this is not clear from the use of 'NSE/FWA'). Page 11 line 54 on sample sizes suggests that there will be no overlap between the types of workers ('Out of these, 2000 will be NSW (sic) and 4000 will have FWAs.') and this should be clarified. Aim1: 'financial viability of the organization, its work environment,', I assume this means their effects on the work environment as experienced by the workers with NSE or FWA (e.g. rather than on all workers in the organization). This seems to relate to RQ2 (?) but it would be useful to clarify this. RQ1: it would be useful to clarify 'opportunities and challenges' for whom (NSE or FWA workers, or other employees or the organization. Financial viability may not be a fixed level or concept (e.g. some NSA's may allow larger surpluses for the organization than other forms of contract etc. and competitors, so how might 'financial viability' be measured in practice?) Normally, reliable financial data is very hard to get from private organisations, but perhaps the situation is different in Sweden (please confirm). In Aim 2 it would be useful to consider more explicitly the views of workers and perhaps their trades unions etc. (the presence of absence of which will be a factor of consideration). Trades unions and employer groups are mentioned later as part of a reference group(s), but my point is about their interaction with workers in or outside the organization. So are unions 'interviewed' as study participants or is their contribution solely as part of the 'advisory'/reference group). Page 7 line 5... Recruitment and Eligibility Criteria Convenience sampling of organizations and employees is to be used, but more details should be given together with some form of framework for seeking to get a suitable range of organizations. For instance, SMEs might be expected to operate differently from larger organizations, and why third sector organizations are to be excluded. Some details should be given on how the potential sample is identified, even if in the end this cannot be fully met. Page 11 line 52: The issue of dropout rates needs to be clarified - are these expected to vary by different groups of workers (within the NSE and FWA groups) and what contingencies are in place?
--	---

	Page 7 Data will be collected c. 2020-2025 – please clarify how long is the delay in getting access to the administrative data from the time it is collected, and how does this match with interview times etc. Please clarify how potential biases in the employee sample recruitment, e.g. due to the HR department acting as ‘gatekeeper initially, are to be avoided. Clarify how potential differences between formal written HR etc. policies and actual implementation by line managers, are to be checked (e.g. are they expected to be done during the interviews with managers as per page 9 line 12?). Page 8, Primary exposures. The issue of hours worked (e.g. different hours for different part-time workers) may be important as well as temporary work. Page 9 lines 6/9: ‘net income from employment as well as total income’, should clarify what is meant by ‘total income’ (e.g. from all sources of employment, from savings interest etc. before/after tax etc.?). Also there is a typographical error in that line. Please clarify how ‘Reduced work performance’ is actually measured, does it apply to subsequent employment and if so how is the comparison calculated? The issue of career progression and training etc. is likely to vary according to NSE as well as FWA and may affect the long term outcomes for the worker and organization in terms of skills development and utilisation. Please clarify how these issues are considered. Page 9 line 20 – there are a range of useful health etc. measurement instruments, but how are they to be applied (is there a danger of respondent fatigue etc.). The roles of personal circumstances, such as childcare, effects of partner, social networks etc. should be considered more fully (e.g. one issue might be that the evidence that an unemployed male partner may lead to lower female employment outcomes). Also the role of the local economy could be more explicitly incorporated (e.g. areas with few alternative job opportunities may have lower turnover). Reasons for the apparent choice to use 5 scale points (page 11 line 48) should be justified. Page 12 line 5: ‘Workers with NSE will be compared with standard workers.....’. How large a sample is expected for standard workers and what data sources will be used to allow full comparisons? Ethics: In relation to page 7 (data holding) I assume there are no data protection issues with processing and holding data in a non-EU country (USA). Similarly I assume that the ethics approval covered issues such as employee data collected not being passed to the employer etc. Can the forms of feedback to the participants who are workers (rather than organizations) be clarified? Will participants be asked about their continued participation at each contact or interview?
--	---

	Clarification should be made of the main ways in which researchers will assist the organizations in improving policies and routines for employment and organization of work and if there is any expected compensation for researchers or their university. Page 5 line 32 and line 41 – is 'NSW' a typographical error or explain difference between work and employment in NSE and NSW. Page 5 lines 36/7 singular/plural grammar corrections needed. "NSE often results in poor physical and psychosocial work environment. However, differences compared to standard work appears small." The figures are not clear (especially Figure 2).
--	---

VERSION 1 – AUTHOR RESPONSE

Reviewer: 1

Prof. Ronald McQuaid, University of Stirling

Comments to the Author:

There are a number of issues that need to be resolved or clarified in the paper.

The project appears to be ready so my comments related mainly to clarifications. The expected/actual start date should be clarified.

General:

This project should add useful research to important areas of understanding. It is important that sufficient resources can be applied to fully achieve the objectives and try to unpack the many confounding influences on employment outcomes.

The expected statistical analysis is outlined in broad terms only (e.g. page 12) and so is difficult to assess at this stage.

The statistical analysis section is, indeed, not detailed when it comes to specific analyses. This is because of the wealth of potential exposures, outcomes, modifiers and confounders, and hence the wide array of statistical methods we will use to analyze associations. Therefore, we would prefer to stay with a broad description to the analyses. This is also necessary considering the word limit of the article.

We do, however, agree with the reviewer that details about data analyses are necessary, but we find this to be more feasible in upcoming studies addressing specific research questions.

The literature is generally appropriate although there is a considerable literature in the social sciences (Economics, Sociology etc.) that would be relevant.

In the introduction, references¹⁻³ from the fields of economics and sociology addressing salary costs, organizational innovation performance, and firm financial performance associated with NSE have been updated or added in the following sentences on page 3: "Some evidence suggests that industries with a large extent of NSE have a weaker product innovation propensity¹, and that NSE is associated with either higher or lower wage costs². Both issues are matters of current debates¹
². Literature addressing effects of hiring NSW on the economic performance of an organization is sparse, although low as well as high proportions of NSE might be negatively associated with financial performance³."

There is considerable policy discussion about 'informal employment' (the informal economy is argued to be regulated, but by informal rather than formal institution, e.g. Williams, 2014) that might be worth mentioning, as some issues may be relevant, although it partly refers to work without contracts, that is excluded from this study (policy reports include ILO reports, European Union's The European Pillar of Social Rights etc.).

We agree that the work environment and health of individuals in informal employment are important and interesting topics for research. We also incorporated the reference to Williams⁴ research in the definition of non-standard work in the article. However, since we are studying several different aspects of flexibility, and as such present several different terms that refer to various aspects of work life flexibility, we found that the introduction of yet another term to describe something that we do not focus in the study lead to the manuscript being less clear. In order to avoid confusion, we would therefore prefer to not incorporate 'informal work' into the study protocol.

Specific managers, and even organizational divisions, may change during the life of the project, so clarify the contingency plans for such events.

When it comes to individuals changing jobs, we have the possibility to follow them in their new position either via their private e-mail and/or in the LISA-register via their social security number. To clarify this, we have revised the description in the "Study procedures" section on page X as follows: "Thus, follow-up data will be available even for study participants that drop out of the study during the data collection period. Participants will also be asked for their private contact information which allows follow-up data to be collected from study participants that have, for instance, changed employer, become unemployed, or turned to studies."

When it comes to changes of organizational divisions, we plan to collect data on structural changes and growth of the investigated departments, and changes in positions or departments for managers and workers, and their turnover. In order to clarify this, we have added the following sentence on page X "This data will include information on organizational structure, including which employees are subordinate to specific managers in each organizational department."

We share the view that different organizational changes will likely occur during the project time. Contingency plans are, to some extent, incorporated in the study design as we have more organizations and respondents than needed as the minimum to conduct the planned studies. This has been described on page X in the "Sample size" section, where we have also revised the following sentence: "This will give enough respondents to cover for dropouts, adjust for relevant covariates and possible effect modification (e.g. by age and sex), in comparisons between categories of workers." On page X, we state that the study population will consist of approximately 10 organizations. With 10 organizations we will still have sufficient data, even in case one organization would cease operation entirely during the project time.

What are dimensions of organizational sustainability and what effects it (which may vary by the sector it is in)? Clarification is needed on the differences between organizational 'sustainability' (e.g. page 5 line 24) and 'viability' (page 6 line 6) and 'financial viability' (page 6 line 13) and more generally what is meant by 'organizational sustainability'. For example, how do they differ?

Also the issue of sustainability for NSE should be clarified.

We thank the reviewer for pointing out the need for clarifying sustainability and its operationalization. We realize that the terminology regarding organizational viability needs clarification. To clarify the scope of the project, we have changed the terminology from "organizational sustainability" to "economic and social sustainability". To further clarify the issue of economic sustainability at an organizational level, we have further changed the terminology from "viability" to "performance". We hope that these revisions make it more clear that our interest is in

three levels of sustainability: individual (individual health outcomes), social (balance between the needs of different organizational stakeholders) and economic sustainability (financial performance of organizations).

The issue of sustainability for NSE is related to the social dimension of sustainability. As described on page 2, social and economic sustainability can be obtained only in a tradeoff between the needs of different stakeholders in organizations. To clarify this, we have, on page X, changed “negative aspects of flexible work” in the last sentence in the second paragraph in the introduction to “Thus, understanding the tradeoff between positive and negative aspects of flexibility in working life is a pertinent issue.” We hope that this revision will make it clearer that this paragraph addresses both NSE and FWA.

References should be given to link and expand on the section including “aiming at determining potential temporal associations between NSE and FWA on the one hand, and organizational viability, the work environment, and health and well-being of the workers, on the other.” (page 6 lines 6/7). We have added new references (Choudhury et al. ⁵, Ciarniene et al. ⁶, Emerson et al. ⁷ and Monteiro et al. ⁸) to back up this argument.

In terms of the aims, it could be made explicit that NSE and FWA may affect different workers but might affect the same worker (as this is not clear from the use of ‘NSE/FWA’). Page 11 line 54 on sample sizes suggests that there will be no overlap between the types of workers (‘Out of these, 2000 will be NSW (sic) and 4000 will have FWAs.’) and this should be clarified. This issue has been clarified by adding the following sentence on page X: “In some cases, a particular worker may be represented in both the latter categories.”

Aim1: ‘financial viability of the organization, its work environment,’, I assume this means their effects on the work environment as experienced by the workers with NSE or FWA (e.g. rather than on all workers in the organization). This seems to relate to RQ2 (?) but it would be useful to clarify this. We have an interest in the work environment for both standard and non-standard workers. To clarify this, we have rephrased the text preceding aim 1. The text now reads “Few studies have compared the working conditions of NSW to those of standard workers.” As regards FWA, studies indicate that the proportion of FWA in an organization affect the working conditions of colleague workers with non-FWA. We have added the following sentence this in the section on work environment and health in flexible work on page X: “The proportion of FWA in an organization may influence the working conditions even for workers with non-FWA⁹.” We hope that these revisions will make aim 1 clearer. RQ1: it would be useful to clarify ‘opportunities and challenges’ for whom (NSE or FWA workers, or other employees or the organization. Financial viability may not be a fixed level or concept (e.g. some NSA’s may allow larger surpluses for the organization than other forms of contract etc. and competitors, so how might ‘financial viability’ be measured in practice?)

Research question 1 concerns aspects regarding challenges for companies / organizations in terms of their financial performance, management accounting and management control. We realize that this can vary with NSE / FWA in different types of organizations. We have therefore rephrased the issue for greater clarity: “What are the opportunities and challenges presented by NSE/FWA in the context of an organization’s financial performance and its management accounting and control?”

Normally, reliable financial data is very hard to get from private organisations, but perhaps the situation is different in Sweden (please confirm).

We have an agreement with the involved companies and organizations to receive financial data at the department and organizational levels. Our experience is that it is feasible to obtain this data directly from company records, in collaboration with the companies, and we expect data to be reliable. For municipalities and affiliated enterprises, all records are public and should be provided upon request within a reasonable time, according to national laws.

In Aim 2 it would be useful to consider more explicitly the views of workers and perhaps their trades unions etc. (the presence of absence of which will be a factor of consideration). Trades unions and employer groups are mentioned later as part of a reference group(s), but my point is about their interaction with workers in or outside the organization. So are unions 'interviewed' as study participants or is their contribution solely as part of the 'advisory'/reference group). We agree that the perspectives of workers and union representatives would be valuable when it comes to evaluation of policies and routines concerning NSE/FWA. These matters are in part covered by the survey (FWA policies) and will also be addressed in interviews (FWA and NSE policies). We have clarified this in RQ6: "How knowledgeable are managers, workers and union representatives regarding policies and routines for NSE and FWA?" and we have also added that "Interviews with managers, workers and union representatives will offer information on the knowledge of NSE/FWA policies and of the extent to which they are implemented" on page 7-8.

VERSION 2 – REVIEW

REVIEWER	McQuaid, Ronald University of Stirling, Management Work and Organisation
REVIEW RETURNED	27-Apr-2022
GENERAL COMMENTS	The responses to my earlier comments have resulted in a clearer paper I think. A few small issues were not explicitly considered (e.g. mention of Third sector organisations) but the authors' replies are sufficient. It should be a very interesting study. A small typo in one reply is: "Participating organizations will receive information that can be tracked down to individual workers." This omits the work 'not' but the correct version is included in the revised article text and in a later reply also.